# The Regulatory Mechanism of Transthyretin Irreversible Aggregation through Liquid-to-Solid Phase Transition

**DOI:** 10.3390/ijms24043729

**Published:** 2023-02-13

**Authors:** Guangfei Duan, Yanqin Li, Meimei Ye, Hexin Liu, Ning Wang, Shizhong Luo

**Affiliations:** Beijing Key Laboratory of Bioprocess, College of Life Science and Technology, Beijing University of Chemical Technology, Beijing 100029, China

**Keywords:** transthyretin, liquid-liquid phase separation, aggregation

## Abstract

Transthyretin (TTR) aggregation and amyloid formation are associated with several ATTR diseases, such as senile systemic amyloidosis (SSA) and familial amyloid polyneuropathy (FAP). However, the mechanism that triggers the initial pathologic aggregation process of TTR remains largely elusive. Lately, increasing evidence has suggested that many proteins associated with neurodegenerative diseases undergo liquid–liquid phase separation (LLPS) and subsequent liquid-to-solid phase transition before the formation of amyloid fibrils. Here, we demonstrate that electrostatic interactions mediate LLPS of TTR, followed by a liquid-solid phase transition, and eventually the formation of amyloid fibrils under a mildly acidic pH in vitro. Furthermore, pathogenic mutations (V30M, R34T, and K35T) of TTR and heparin promote the process of phase transition and facilitate the formation of fibrillar aggregates. In addition, S-cysteinylation, which is a kind of post-translational modification of TTR, reduces the kinetic stability of TTR and increases the propensity for aggregation, while another modification, S-sulfonation, stabilizes the TTR tetramer and reduces the aggregation rate. Once TTR was S-cysteinylated or S-sulfonated, they dramatically underwent the process of phase transition, providing a foundation for post-translational modifications that could modulate TTR LLPS in the context of pathological interactions. These novel findings reveal molecular insights into the mechanism of TTR from initial LLPS and subsequent liquid-to-solid phase transition to amyloid fibrils, providing a new dimension for ATTR therapy.

## 1. Introduction

Amyloid protein aggregates, such as FUS and TDP-43 aggregations in Amyotrophic Lateral Sclerosis (ALS), Tau and Aβ aggregations in Alzheimer’s disease (AD), and α-synuclein fibrils in Parkinson’s disease (PD), are pathognomonic features of several neurodegenerative diseases [1,2]. Several studies have found that liquid-liquid phase separation (LLPS) can cause protein pathological aggregation in disease [3,4,5,6]. Weak multivalent interactions, such as electrostatic, cation-π, π-π, and dipole-dipole interactions, drive LLPS [2,7,8,9,10,11,12,13,14]. Later, the droplets formed by LLPS become more viscoelastic and rigid over time and eventually lose the ability to exchange internal component molecules with their surroundings [1,3,4,15,16,17], which has been termed “in vitro aging” [3]. The liquid-to-solid phase transition occurs due to the entanglement of biopolymers or stronger interactions, which might cause the formation of amyloid-like fibrils and accelerate the nucleation rate [1,3,4,15,17,18]. This liquid-to-solid phase transition is widely accepted as the basis for intracellular pathological proteins fibrillar formation in neurodegenerative diseases. The irreversible condensates solidification is due to key mutations, interaction partners [19] and unstable physicochemical conditions (osmotic pressure, salt concentration, temperature, pH, etc.) [20,21]. However, post-translational modifications (PTMs) can also alter phase transition and aggregation of proteins by modulating protein charge, steric properties and interaction strength [22,23,24,25,26,27], thus regulating the characteristics and overall function of the condensate. Evidence suggests that PTMs [28] such as phosphorylation [29,30], methylation [31,32], acetylation [33,34,35,36], and others influence phase transition behavior. Exploring the molecular mechanisms of protein liquid-to-solid phase transitions and aggregation is fundamental to understanding the onset of disease processes.

Transthyretin (TTR) is a 55-kDa homo-tetrameric protein with a globular structure, composed of four identical subunits [37,38,39]. Each monomer is made up of eight antiparallel β-sheets (A-H) and one short α-helix [37]. Inner β-sheets are formed by the interaction of hydrogen bonds between A, D, G and H strands, whereas outer β-sheets are formed by B, C, E and F strands. When two TTR monomers come together, they form a dimer with two twisted eight-stranded β-sheets. Two dimers associate through a dimer-dimer interface to form a tetrameric structure. TTR is primarily synthesized in the liver [40,41], brain choroid plexus [42], retinal pigment epithelium and pancreas, excreting into plasma and cerebrospinal fluid, respectively. TTR has the ability to transport thyroid hormone and retinol in plasma and CSF, as well as proteolytic activity; as a result, it plays critical roles in a variety of processes [43]. Senile systemic amyloidosis (SSA) is associated with age-onset extracellular deposition of wild-type TTR in cardiac tissue [44,45]. More than 100 amyloidogenic TTR mutants lead to the formation of amyloidosis [46] such as familial amyloid polyneuropathy (FAP) [47], familial amyloid cardiomyopathy (FAC) [48] and central nervous system-associated amyloidosis (CNSA) [49,50,51], mainly depositing in the heart, lungs and peripheral nerves [52]. Previous studies have speculated that TTR misfolding and aggregation may be triggered in an acidic environment such as endosomes and lysosomes in vivo [53]. Mechanistic studies have shown that TTR forms amyloid in vitro through a pH-mediated denaturation process [54,55] and mildly acidic pH 4–5 is the optimum conditions [54]. As the pH decreases, the native TTR tetrameric structure rearranges and disassociates into monomers, and then self-assembles into amyloid fibrils [55,56]. In addition, TTR is frequently modified with many different compounds at Cys10 by forming disulfide bonds [57,58,59], which alters the propensity of TTR amyloidosis by modulating TTR stability. S-sulfonation and S-cysteinylation usually occur in plasma and CSF [60,61,62,63]. Although the aggregation of TTR mutants and PTMs at mildly acidic levels has received more attention, the triggering mechanism of the TTR aggregation process has not been explored.

In this study, we explored the mechanism of TTR LLPS and its conversion from liquid-to-solid phase transition to aggregates under mildly acidic conditions. We further revealed that pathogenic mutations (V30M, R34T, and K35T) accelerate phase transitions and facilitate the formation of fibrillary aggregates. In addition, we demonstrated that S-cysteinylation increased the aggregation propensity and S-sulfonation reduced the aggregation rate, which were mediated through the phase transition process. These results showed TTR LLPS and the subsequent liquid-to-solid phase transition, which may provide a new perspective for understanding the formation of TTR aggregates in SSA and FAP.

## 2. Results

### 2.1. The Phase Separation of TTR Was Affected by pH

TTR tetramer disassociates into misfolded monomer and self-assembles into amyloid fibrils via a pH-mediated denaturation process in vitro [54,55,56]. Recent research has shown that LLPS can cause pathological protein aggregation in neurodegenerative disease [3,4,5,6]. To investigate the initial step of TTR pathological aggregation, we explored the behavior of LLPS in response to acidic conditions. In this study, TTR with an enhanced green fluorescent protein (EGFP) fusion label at the N-terminus was expressed in *Escherichia coli* and purified by metal affinity chromatography. To explore the effect of EGFP tag on the formation of TTR tetramers, the molecular weight of EGFP-TTR was identified by native-PAGE. The molecular weight of EGFP-TTR was about 160 kDa, which proved that EGFP-TTR could form tetramers (Appendix A). The status of EGFP-TTR at various acidic pHs was explored under a confocal microscope. Protein droplets were formed and enlarged at pH 5.5, indicating the occurrence of LLPS. As the pH was further decreased, both the size and number of droplets increased (Figure 1A). A small amount of irregular aggregation existed around pH 6 (Figure 1A), indicating that EGFP-TTR was in a non-stabilized tetramer state [56]. The turbidity of the solution gradually increased with the decrease in pH, which was consistent with the image-based analysis (Figure 1B). To rule out the effect of the EGFP tag, TTR was stained with Cyanine 3 (Cy3), which revealed that TTR also formed phase separation (Figure 1A). We also compared the EGFP alone, and under the same conditions, the EGFP alone did not form droplets (Appendix A). Considering the highly charged properties of TTR, we assumed that the LLPS was driven by electrostatic interactions. We monitored the phase separation of EGFP-TTR under different salt concentrations (Figure 1C and Appendix A) and found that the LLPS was inhibited at high salt concentrations. These finding confirmed that TTR formed phase separation due to electrostatic interactions at acidic pH.

### 2.2. TTR Droplets Mature and Age into Fibrillar Aggregates

Proteins associated with neurodegenerative diseases, such as FUS, Tau, TDP-43 and α-Synuclein, can transition from liquid droplets to solid droplets and then into amyloid-like or amorphous protein aggregates [3,4,64,65,66]. Endosomes/lysosomes pH was used to induce TTR tetramer dissociation (the rate-limiting step) and trigger the conformational rearrangement for amyloid fibril formation, promoting TTR amyloid formation [67]. At various time intervals, we observed the entire TTR transition and aggregation process using Thioflavin T (ThT) staining and transmission electron microscopy (TEM) (Figure 2A). The confocal microscope showed that TTR gradually matured and aged from droplets to aggregated fibers (Figure 2A). Next, we investigated the effect of EGFP tag on TTR transition and aggregation process and the factors of TTR fibril aggregation. We used ultra-centrifugation to eliminate the dilute phase (monomers and oligomers) and then used the dense phase (droplets) to measure liquid-to-solid transition. The dense phase of EGFP-TTR also showed the same gradual maturation and aged from droplets to aggregated fibers (Appendix A). This also proved that TTR fibrils were formed through liquid-to-solid phase transition. We performed fluorescence recovery after photobleaching (FRAP) experiments to evaluate the liquid-to-solid phase transition, which was characterized by molecular diffusion transformation. The recovery of bleached fresh EGFP-TTR droplets reached 60% (Figure 2B) but decreased (about 20%) rapidly after 1 h of incubation (Figure 2C), which indicated that TTR liquid droplets rapidly matured into gels with high viscosity. Droplets fusion experiments also demonstrated the fluidity of fresh EGFP-TTR droplets (Figure 2D). TTR rapid polymerization was revealed by ThT fluorescence kinetics curve, indicating the presence of amyloid-like β-sheet structures (Figure 2E). TEM imaging further confirmed the formation of amorphous aggregates over time (15 days) and their eventual conversion to amyloid fibrils (30 days) (Figure 2A).

The small molecule inhibitor tafamidis could effectively stabilize the TTR teramer and prevent depolymerization into monomers [68,69]. We found that tafamidis was able to effectively inhibit the formation of the TTR phase separation under acidic conditions (Figure 2F). This further proved that TTR phase separation was formed by TTR monomers. The ThT fluorescence kinetic curve also revealed that tafamidis was able to inhibit the polymerization of TTR, indicating that it was stable in tetramers (Figure 2G). Heparan sulfate (HS) was detected as a pertinent component in the heart of cardiomyopathy associated with TTR amyloid deposition by immunohistochemical staining [70,71,72]. Thus, we tested whether heparin (an analogue of HS) had an effect on TTR LLPS and aggregation. We confirmed that heparin promoted phase separation at low concentrations and inhibited it at high concentrations by turbidity and confocal microscope (Figure 2H and Appendix A). FRAP analysis of droplets showed that the fluorescence recovery rate of droplets was 60% when the molar ratio of TTR to heparin was 1:1, while the droplet did not recover when the molar ratio of TTR to heparin was 1:10 (Figure 2I), which indicated that more heparin led to early maturation of the droplets and eventually caused enhanced aggregation. Indeed, previous studies found that heparin reinforced droplets interaction networks to promote LLPS at a low concentration, but began to repulse and thus suppress its initial effect at a high concentration [73]. Furthermore, the ThT fluorescence kinetics curve revealed that heparin promoted TTR fibrillogenesis (Figure 2J). TEM imaging further confirmed the formation of amyloid-like fibrils (Figure 2K). In conclusion, TTR droplets mature and age into fibrillar aggregates, and heparin accelerates the progress from initial LLPS to aggregation. Tafamidis inhibits TTR depolymerization to form droplets and thus inhibits aging fibroblast aggregates. This indicates that TTR depolymerization is necessary for phase separation and amyloid production.

### 2.3. Disease-Associated Mutations Accelerate TTR Droplets Maturation and Aging into Fibrils

Previous studies have shown that disease-associated mutations of FUS might induce droplet formation as an aggregation precursor and accelerated droplets solidification and fibrils formation in ALS/FTD [16]. Therefore, we raised the question of whether disease-associated mutations alter the propensity of TTR to undergo LLPS or induce transformation into aggregates. The V30M mutation was the most common mutation in FAP. The R34T mutation caused vitreous amyloidosis [74]. The K35T mutation was identified in patients with amyloid polyneuropathy and restrictive cardiomyopathy [75]. To probe this issue, we analyzed the effects of disease-associated mutations (V30M, R34T, and K35T) on TTR phase separation and aggregation. The confocal microscope imaging showed that EGFP-TTR (V30M) and EGFP-TTR (R34T) mutants formed more droplets than EGFP-TTR (Figure 3A and Appendix A). In contrast, EGFP-TTR (K35T) formed non-spherical aggregates (Figure 3A and Appendix A). FRAP analysis of droplets showed that the dynamics of the droplets formed by EGFP-TTR (V30M) and EGFP-TTR (K35T) were rapidly lost, and EGFP-TTR (R34T) droplets solidified more slowly than EGFP-TTR. This indicated that mutants accelerated the process of liquid-to-solid phase transition (Figure 3B). Furthermore, the ThT fluorescence kinetics curve illustrated that the stability extent decreased for all three TTR mutants even through the level of reduction was variable: the reduction was the highest for K35T; intermediate was V30M; and the lowest reduction was R34T (Figure 3C). TEM imaging further confirmed that K35T formed more amyloid fibrils than the other two mutants (V30M and R34T) (Figure 3D). In summary, our data demonstrated that phase separation was a crucial step prior to the initiation of TTR aggregation, and disease-associated mutants (V30M, R34T, and K35T) accelerated the process from liquid-to-solid phase transition to amyloid fibrils formation.

### 2.4. The Effects of S-cysteinylation and S-sulfonation on TTR Droplets and Aggregation

PTMs of Cys_10_ cysteine residue play crucial roles in TTR aggregation in hereditary transthyretin amyloidosis diseases [58,76,77,78]. S-cysteinylation and S-sulfonation are the most common modifications in the blood and cerebrospinal fluid, which affect TTR stability and alter the propensity of TTR to form amyloid fibrils [79,80,81,82,83]. In order to study the effect of these two PTMs on TTR aggregation, we synthesized these two modified TTR proteins. According to previous modification methods [79,81], we successfully obtained the S-cysteinylated TTR (15,944.911 Da, △+119) and S-sulfonated TTR (15,907.755 Da, △+80), which were verified using a MALDI-TOF mass spectrometer (Appendix A). To prove our conjecture that aggregation was indeed achieved through phase transition, we compared the phase separation and aggregation abilities of S-cysteinylation and S-sulfonation for TTR. Importantly, we found that S-cysteinylated and S-sulfonated of EGFP-TTR could form spherical droplets detected by confocal microscope under identical mildly acidic pH (Figure 4A), whereas FRAP experiments revealed no mobility of these droplets (Figure 4B,C), indicating that phase separation could occur early and result in rapid maturation of the droplets. Then, we used ThT fluorescence kinetics to explore the effect of modification on TTR aggregation. Consistent with previously published data [81,84], the ThT fluorescence kinetics curve showed that S-cysteinylation was more effective than S-sulfonation in increasing the propensity for TTR aggregation (Figure 4D), and TEM imaging showed that S-cysteinylated TTR formed amyloid fibrils and S-sulfonated TTR formed oligomers (Figure 4E). Similarly, the stability extent of S-sulfonation and aggregation propensity of S-cysteinylation for mutants (V30M, R34T, and K35T) were more effective than unmodified TTR mutants (Figure 4F,G and Appendix A), which all formed condensates with no mobility (Appendix A). Taken together, S-cysteinylation increased the aggregation propensity and S-sulfonation stabilized tetramer structure for TTR and mutations (V30M, R34T, and K35T), which quickly experienced the process of phase transition.

## 3. Discussion

A series of studies indicated that many proteins that tend to aggregate can undergo LLPS in neurodegenerative diseases, and the liquid-to-solid transition is also associated with amyloid fibrils during the pathogenic process [85,86,87]. Although TTR misfolding, aggregation structure and kinetics have been extensively studied, the triggering mechanism for the initial aggregation of TTR remains unclear. In this study, we found that TTR underwent phase transition from an initial liquid droplet state to a gel-like state of high viscosity and finally to amyloid fibrils in vitro through applying multiple biophysical techniques, including the confocal microscope, FRAP, ThT, and TEM.

Lysosome/endosome pH was known to induce TTR tetramer dissociation (the rate-limiting step) and trigger the conformational arrangement for amyloid fibrils formation in vivo [54]. Along this line, TTR formed phase separation under mildly acidic pH, and the addition of salt suppressed droplets formation (Figure 1), which emphasized the importance of electrostatic interactions for droplets formation. Indeed, solution and solid-state NMR spectroscopy, molecular dynamics simulations, and hydrogen exchange experiments showed that A and G strands of TTR converted from β-sheet to α-sheet and form two complementary charged interfaces through peptide backbone conformations [88,89,90,91,92]. The result may provide an electrostatic driving force for TTR to undergo LLPS (Figure 5). Tafamidis is a stabilizer of TTR natural tetramer structure for the treatment of TTR amyloid-related diseases. Our study demonstrated that tafamidis was able to inhibit droplet formation by stabilizing TTR tetramers, thereby inhibiting amyloid production. It provided theoretical support for understanding the mechanism of treatment of related diseases. Furthermore, heparin had opposite effects on TTR phase separation at low and high concentrations, possibly because negatively charged heparin interacted with TTR peptide motifs (31-HVFRK-35) through electrostatic interaction at low concentrations [72,93], while high concentrations of heparin began to reject, so it was inhibited at high concentrations [73]. Then, because of the strong β-sheet structure, heparin accelerated droplets solidification, promoting the formation of amyloid fibrils.

Several studies [69,94,95] have demonstrated the mechanism that TTR and V30M mutation aggregated and amyloidogenic fibrillation propensity under acidic conditions. Acidic induced TTR to unfold towards the intermediate state of monomers that were prone to aggregation, and V30M mutation accelerated this process. Our results also showed that V30M mutation was easier to phase separate than TTR under acidic pH, and improved the molecular mechanism of TTR phase separation and aggregation. Since disease-associated mutations form amyloid fibrils at a higher pH compared with WT [46,55,96,97,98,99], pH 5.5 was selected to study the differences in mutations. Three pathogenic mutations (V30M, R34T, and K35T) accelerated the liquid-to-solid phase transition and facilitated the formation of fibrillar aggregates (Figure 3), which might be due to structural changes caused by the mutation. However, the question still remained concerning why there were significant differences in aggregation rates among the three pathogenic mutations. One answer could be that Lys-35 acted as a gatekeeper residue to diminish the aggregation propensity of the aggregation-prone region 26–57 by sequence-specific [100], and thus TTR V30M and R34T mutations were not highly aggregated owing to Lys-35 residue protection. Depending on the position of the modified amino acid residue, PTMs can also affect phase transition and aggregation of wild-type and mutant proteins. Like mutations, S-cysteinylation reduced the kinetic stability of TTR and increased the propensity for aggregation because the bulky Cys group caused steric interference, which accelerated the process from phase transition to aggregation [79]. S-sulfonation stabilized TTR tetramer by two sulfite oxygen, providing extra intramolecular hydrogen bond interactions between residues in strand A and in the vicinity of D [82], while disrupting the local structure that maintained two complementary charged interfaces between A and G strands, which accelerated liquid-to-solid transition and reduced aggregation rate (Figure 4).

Based on the above observations, we propose a model for LLPS-mediated TTR aggregation shown in Figure 5. Under mildly acidic pH, TTR undergoes LLPS through electrostatic interaction and a subsequent liquid-to-solid phase transition, as well as amyloid fibril formation in vitro, which is accelerated by pathogenic mutations (V30M, R34T, K35T) and heparin. In addition, S-cysteinylation destabilizes the TTR tetramer because the bulky Cys group caused steric interference to promote amyloid aggregation, while S-sulfonation stabilizes TTR tetramer by extra intramolecular hydrogen bond interactions to decrease aggregation, both of which undergo a phase transition process. A detailed elucidation of the mechanisms of initial and progressive aggregation in FAP/SSA diseases is crucial not only for understanding TTR pathogenic process, but also for developing a new ATTR therapeutic approach that disrupts pathogenic progression in the stage of LLPS. Indeed, accumulating evidence indicates that small molecules can disrupt aberrant phase transitions while maintaining physiological function [101,102,103].

## 4. Materials and Methods

### 4.1. Recombinant Protein Expression and Purification in Escherichia coli

TTR and mutants including V30M, R34T and K35T were cloned into the pET-28a and pET-28a-EGFP vector (Miao Ling Plasmid, Wuhan, China). Finally, TTR with EGFP at the N-terminal were constructed. All proteins were expressed in *Escherichia coli* (BL21 (DE3), TransGen Biotech, Beijing, China). Bacteria were grown at 37 °C until OD_600_ reached 0.8–1.0, then induced with 0.5 mM isopropyl-β-d-thiogalactopyranoside (IPTG) at 25 °C overnight. The cells were centrifuged and lysed. The protein was purified by a Ni-NTA metal affinity column (GE Healthcare). TTR solutions were stored at −80 °C. TTR purity was identified by SDS-PAGE with Coomassie blue staining.

### 4.2. TTR Oxidative Modification

The recombinant protein was desalted to filtered phosphate buffer (10 mM sodium phosphate, 100 mM KCl, and 1 mM EDTA, pH 7.0) using a desalting column. Next, 400 μM cysteine or 40 mM sodium tetrathionate dissolved to filtered phosphate buffer. To get the S-cysteinylation modification on Cys10 of TTR, we incubated an equal volume of TTR with 400 μM cysteine at 25 °C, 50 rpm, 15 h. To get the S-sulfonation modification on the same residue, we incubated an equal volume of TTR with 40 mM sodium tetrathionate at 25 °C, 50 rpm, 15 h. Reactions were terminated by desalting into phosphate buffer, and the modified proteins were stored at −80 °C.

### 4.3. Phase Separation of EGFP-TTR under Different Acidic Conditions

All purified and modified EGFP-TTR were prepared by desalting using KMEI buffer (50 mM KMEI, 1 mM MgCl_2_, 1 mM EGTA, and 10 mM imidazole, pH 7.0) through a ÄKTA pure machine (General Electric, Schenectady, NY, USA) for the next-step phase separation studies.

To induce TTR condensates formation at acidic conditions, EGFP-TTR in KMEI buffer (pH 7.0) was mixed with equal volumes of acetate KMEI buffer to the desired pH (4.4 or 5.5) at a final concentration of 10 μM. Mutants (EGFP-TTR V30M, EGFP-TTR R34T and EGFP-TTR K35T) and the modified proteins (S-cysteinylated TTR and S-sulfonated TTR) were performed the same way.

To study the effect of heparin on phase separation, heparin was dissolved in KMEI buffer (pH 7.0) and mixed with 10 μM EGFP-TTR (pH 7.0) at molar ratios of 1:0, 1:1, 1:5, 1:10, 1:50 (EGFP-TTR: heparin). Then, an acidic KMEI buffer was added to adjust pH.

### 4.4. Cy3 Staining

Next, 10 mg/mL Cy3-NHS ester (dissolved in DMSO) was prepared. Then, 7 μL Cy3 dye (10 mg/mL) and 5 μL NaHCO_3_ (1 M) were mixed with 93 μL TTR protein (100 μM) in KMEI buffer (pH 7.0). The mixture was incubated for 1 h at 37 °C, 220 rpm. The excess Cy3 dye was removed by Slide-A-Lyzer MINI dialysis devices (10K MWCO, 0.1 mL, Thermo Scientific, Waltham, MA, USA) against KMEI buffer (pH 7.0). Unlabeled TTR protein was mixed with Cy3 dye at a molar ratio of 20:1 for confocal experiments.

### 4.5. Turbidity Assay

Turbidity (optical density at 600 nm) was measured in 384-well plates using a SpectraMax M2 microplate reader. All samples were examined in triplicate (n = 3). Results were analyzed by GraphPad Prism 7.0.

### 4.6. Confocal Microscope

The samples were immediately loaded into a 96-well plate and imaged with the help of a Leica SP8 microscope with a 100× oil immersion objective. The droplets were observed with an EGFP (448 nm) fluorescence channel. ImageJ was used to identify droplets and analyze their number, area and circularity. Results shown are representative of at least three biological replicates.

### 4.7. Fluorescence Recovery after Photobleaching (FRAP)

FRAP measurements were performed on an inverted LSM 780 microscope (Observer.Z1; Carl Zeiss, Oberkochen, Germany) with a Zeiss 100× oil immersion lens and a confocal spinning disk unit (CSU-X1; Yokogawa, Tokyo, Japan). A 488-nm laser [1 AU (Airy Unit)] with 100% laser power was used to bleach the droplets. Post-bleach images were collected at a rate of 1 s per frame for 100 s. Fluorescence intensity was normalized with pre-bleach as 100% and post-bleach as 0. Results (at least three FRAP curves) were analyzed by GraphPad prism 7.0.

### 4.8. Thioflavin T (ThT) Staining

The florescence of 10 μM TTR mixed with 20 μM ThT in 150 mM KMEI buffer (pH 4.4) was visualized using a Leica SP8 microscope with 514 nm excitation and ~527 nm emission channel (yellow) at different time points. The results were analyzed by GraphPad Prism 7.0.

### 4.9. Thioflavin T (ThT) Binding Assay

All the prepared TTR proteins in phosphate buffer (10 mM sodium phosphate, 100 mM KCl, and 1 mM EDTA, pH 7.0) were mixed with equal volumes of acetate buffer (200 mM NaAc, 100 mM KCl, 1 mM EDTA) to a final concentration of 40 μM and the desired pH (4.4 or 5.5) to initiate reactions. The solutions with 20 μM ThT dye were added to a 96-well black plate. Aggregate kinetics were monitored by ThT fluorescence on the SpectraMax M2 microplate reader with excitation and emission wavelengths of 440 and 485 nm, respectively. Measurements were taken every 1 h at 37 °C for 5 days. The aggregate rates of heparin, TTR were measured in the same way. All samples were examined in triplicate (n = 3).

### 4.10. Transmission Electron Microscopy (TEM)

Samples were prepared in the same manner as for the ThT assay and incubated at 37 °C for 1–2 months. Then, 10 μL of sample was spotted onto carbon film-coated 400-mesh copper grids for 4 min, and then stained with 2% sodium phosphotungstic acid aqueous solution (pH 6.5) for 2 min. Images were taken using an Hitachi HT7700 transmission electron microscope (Shimomatsu City, Japan) at 120 kV.

### 4.11. Evaluation of MALDI-TOF-MS Spectra

The MALDI-TOF mass spectrometer (BRUKER, Ultraflex Treme, Karlsruhe, Germany) was used for evaluating the molecular weights of all samples. The successful oxidation modification masses of TTR, S-cysteinylated TTR and S-sulfonated TTR were 15,826.161 Da, 15,944.911 Da (△ = 119), and 15,907.755 Da (△ = 80). From the mass spectra, we estimated that the reaction was complete.

## Figures and Tables

**Figure 1 ijms-24-03729-f001:**
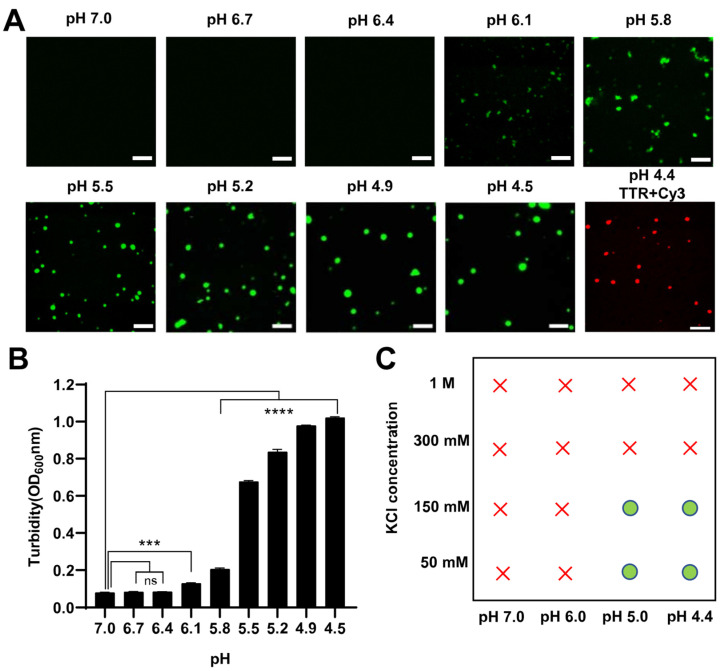
TTR forms phase separation under acidic conditions. (**A**) Confocal microscope of 10 μM EGFP-TTR at various pHs and Cy3 staining of 10 μM TTR at pH 4.4. Scale bar, 2.5 μm. (**B**) Turbidity of 10 μM EGFP-TTR solution at various pHs. Error bars represented SEM (n = 3). **** *p* ≤ 0.0001, *** *p* ≤ 0.001, ** *p* ≤ 0.01, * *p* ≤ 0.05. The *p* values for 10 μM EGFP-TTR at pH 6.7 (*p* = 0.9996), pH 6.4 (*p* = 0.9994), pH 6.1 (*** *p* = 0.0002), pH 5.8 (**** *p* < 0.0001), pH 5.5 (**** *p* < 0.0001), pH 5.2 (**** *p* < 0.0001), pH 4.9 (**** *p* < 0.0001) and pH 4.5 (**** *p* < 0.0001) were calculated with respect to pH = 7.0. (**C**) The regime diagram illustrated the phase separation of 10 μM EGFP-TTR at varying pH and salt concentrations. The green dot indicated droplets. The red “×” indicated no phase separation.

**Figure 2 ijms-24-03729-f002:**
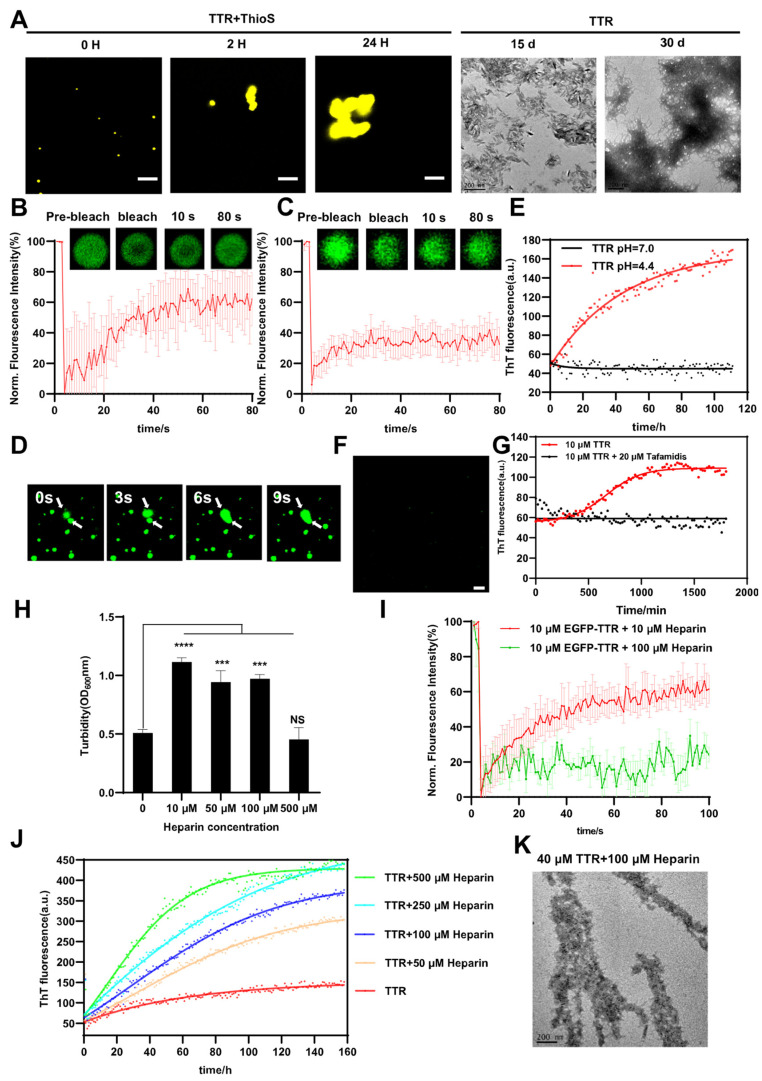
TTR droplets mature and age into fibrillar aggregates, and heparin promotes this process. (**A**) Time-dependent changes of TTR liquid droplets analyzed by ThT staining and TEM imaging. 10 μM TTR were incubated at pH 4.4, 37 °C. Scale bar, 2.5 μm. (**B**) FRAP of 10 μM fresh EGFP-TTR at pH 4.4. Error bars represented SEM (n = 3). (**C**) FRAP of 10 μM EGFP-TTR incubated 1 h at pH 4.4. Error bars represented SEM (n = 3). (**D**) Time-lapse imaging of droplet fusion. 10 μM TTR were incubated at pH 4.4. (**E**) The ThT fluorescence intensity traced for 40 μM TTR at pH 4.4, 37 °C. (**F**) Confocal microscope of 10 μM EGFP-TTR and 20 μm tafamidisat at pH 4.4. Scale bar, 5 μm. (**G**) The ThT fluorescence intensity traced for 10 μM TTR and 20 μM tafamidis at pH 4.4, 37 °C. (**H**) Turbidity of 10 μM EGFP-TTR in the presence of varying heparin concentrations at pH 4.4. Error bars represented SEM (n = 3). **** *p* ≤ 0.0001, *** *p* ≤ 0.001. The *p* values for 10 μM heparin (**** *p* < 0.0001), 50 μM heparin (*** *p* = 0.0008), 100 μM heparin (*** *p* = 0.0005) and 500 μM heparin (NS) were calculated with respect to 10 μM EGFP-TTR. (**I**) FRAP of 10 μM EGFP-TTR in the presence of 10 μM and 100 μM heparin. Error bars represented SEM (n = 3). (**J**) The ThT fluorescence intensity traced for 40 μM TTR in the presence of varying heparin concentrations at pH 4.4, 37 °C. (**K**) TEM imaging of fibrillar aggregates formed from 40 μM TTR in the presence of 100 μM heparin for 30 days at 4.4, 37 °C.

**Figure 3 ijms-24-03729-f003:**
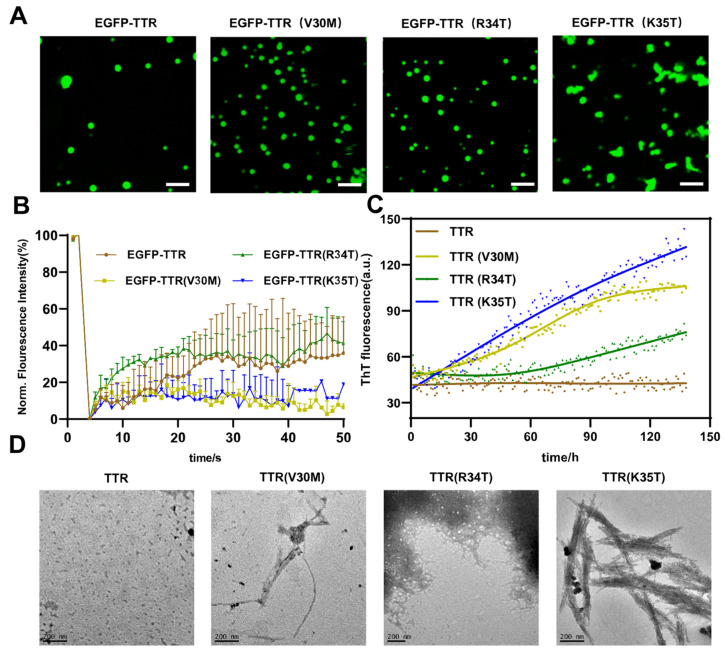
Disease-associated mutations accelerate TTR droplets maturation and aging into fibrils. (**A**) Confocal microscope images of 10 μM EGFP-TTR and disease-associated mutants (V30M, R34T, K35T) with an EGFP fusion label at pH 5.5. Scale bar, 2.5 μm. (**B**) FRAP of 10 μM EGFP-TTR and disease- associated mutants (V30M, R34T, and K35T) at pH 5.5. Error bars represented SEM (n = 3). (**C**) The ThT fluorescence intensity was measured for 40 μM TTR and disease- associated mutants (V30M, R34T, K35T) at pH 5.5, 37 °C. Error bars represented SEM (n = 3). (**D**) TEM imaging of fibrillar aggregates formed from 40 μM TTR and disease- associated mutants (V30M, R34T, and K35T) at pH 5.5, 37 °C. Scale bar, 200 nm.

**Figure 4 ijms-24-03729-f004:**
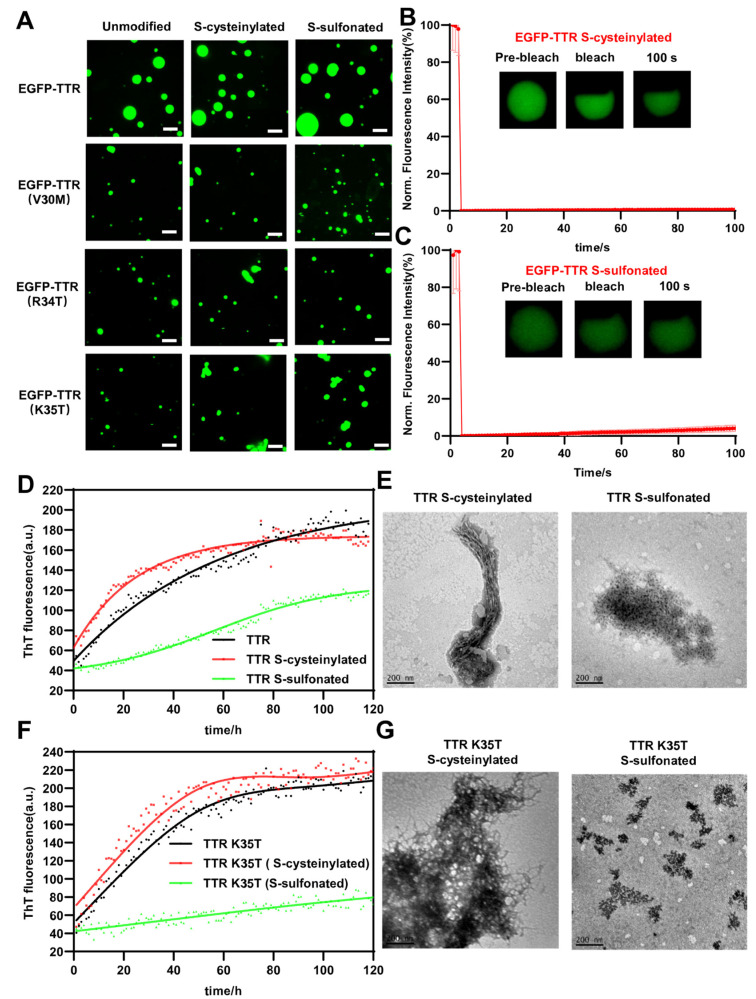
The effects of S-cysteinylation and S-sulfonation on TTR droplets and aggregation. (**A**) Confocal microscope images of 10 μM unmodified, S-cysteinylated and S-sulfonated EGFP-TTRs and disease-related mutants at pH 4.4. Scale bar, 2.5 μm. (**B**) FRAP of 10 μM S-cysteinylated EGFP-TTR at pH 4.4. Error bars represented SEM (n = 3). (**C**) FRAP of 10 μM S-sulfonated EGFP-TTR at pH 4.4. Error bars represented SEM (n = 3). (**D**) The ThT fluorescence intensity traced for 40 μM unmodified, S-cysteinylated and S-sulfonated TTR at pH 4.4, 37 °C. (**E**) TEM image of fibrillar aggregates formed from 40 μM unmodified, S-cysteinylated and S-sulfonated TTR for 30 days at pH 4.4, 37 °C. Scale bar, 200 nm. (**F**) The ThT fluorescence intensity traced for 40 μM unmodified, S-cysteinylated and S-sulfonated TTR (K35T) at pH 4.4, 37 °C. (**G**) TEM image of fibrillar aggregates formed from 40 μM unmodified, S-cysteinylated and S-sulfonated TTR (K35T) for 30 days at pH 4.4, 37 °C. Scale bar, 200 nm.

**Figure 5 ijms-24-03729-f005:**
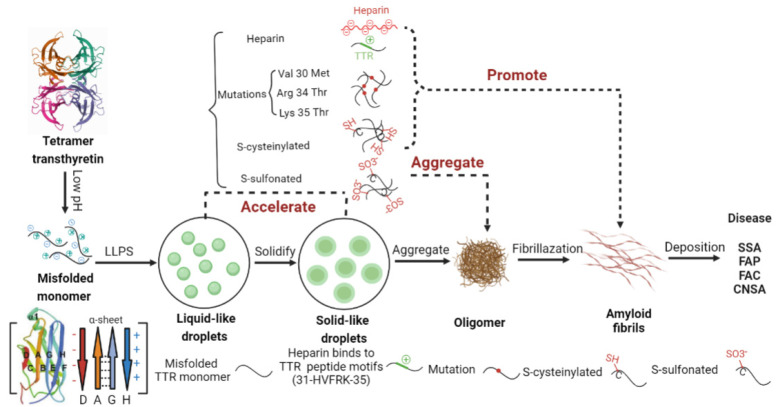
A model for irreversible aggregation of TTR through a liquid-solid phase transition. TTR disassociates into monomers at low pH, then forms oligomers through the liquid-to-solid phase transition, and finally fibrosis to amyloid fibrils that lead to disease. Mutations or PTMs destroy phase separation, leading to the formation oligomer or amyloid fibrils directly.

## Data Availability

Data are available from the corresponding authors upon reasonable request.

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
