# Peer review of "The Regulatory Mechanism of Transthyretin Irreversible Aggregation through Liquid-to-Solid Phase Transition"

_ijms, 2023, doi:10.3390/ijms24043729_

Round 1
Reviewer 1 Report
In this manuscript, the aim of Duan and colleagues was to investigate the phase separation of TTR, as well as its transition mechanism from a liquid-to-solid state under acidic conditions. They demonstrated that pathogenic mutations (V30M, R34T, and K35T) facilitate the formation of aggregation and amyloidogenic fibrillation. Additionally, they showed that S-sulfonation decreased the aggregation rate and S-cysteinylation increased the aggregation propensity, which were both mediated by the phase transition process. It is a well-structured study employing a wide spectrum of techniques including confocal microscopy, FRAP, Turbidity, TEM, and Maldi-TOF MS. However the main research questions and the methodology addressing these questions are suffering major shortcomings and require substantial improvements. Thus, the authors are kindly asked to consider the following major points to improve the manuscript.
1- Considering the current strategy, it is impossible to distinguish whether amyloidogenic TTR fibrils are formed through liquid-to-solid phase transition or by the expansion of oligomers via the use of dilute phase monomers. It is suggested to perform ultra-centrifugation to eliminate the dilute phase (monomers and oligomers), then measure the liquid-to-solid transition of the dense phase.
2- In the liquid phase, it is suggested to show the fusion events of droplets to support the notion that it phase separates.
3- It was previously shown that small molecule stabilizing the tetramer form of TTR inhibits aggregation and amyloidogenic fibril formation (PMID: 12560553). Have the authors tested whether that stabilization also inhibits initial TTR phase separation?
4- The use of the FRAP technique to measure the viscosity - liquidity/solidity of the biomolecule has certain caveats: The recovery of fluorescence relies on several parameters such as diffusion coefficients and concentration of the biomolecule. Even a little change in one of these parameters could cause a change in the recovery rate. For example, the pre-bleach images in Fig.2B-C show large intensity differences between the conditions suggesting potential concentration differences in those individual droplets between the two conditions. Thus, to overcome this challenge, the authors are encouraged to use a more reliable technique such as microrheology to measure the viscosity.
5- There are three essential papers previously published investigating the aggregation of TTR (PMID: 12560553; PMID: 26459562; PMID: 30366153). Even, two of them already investigated the functional importance of the V30M mutation and acidic pH in the aggregation and amyloidogenic fibrillation propensity of TTR, respectively (PMID: 12560553; PMID: 30366153). However, none of these studies are discussed in light of current findings.
Reviewer 2 Report
The paper “The regulatory mechanism of transthyretin irreversible aggregation through liquid-to-solid phase transition” by Duan et al. proposes a mechanism of formation of TTR fibers based on the evidence that, before forming amyloid fibers, TTR aggregates in liquid-like and solid-like droplets.
All experiments have been carried out in vitro at acidic pH, a condition that accelerates a process that at neutral pH takes a much, much longer time.
The major aspect that worries me in data interpretation is the presence of EGFP, used to detect fluorescence. EGFP is a protein of 241 a.a., larger than the TTR monomer (125 a.a). In the Materials and Methods Section, nothing is mentioned about where EGFP was cloned. At the N- or C-terminus of the protein? How long is the spacer between the two? I imagine every monomer of TTR contains one EGFP. How does the presence of such a larger protein influence the stability of the tetramer (and of the monomer)? Is the time necessary for the formation of droplets similar to that necessary in the case of wild-type TTR at the same pH?
In order to form fibers, TTR must undergoes the tetramer-monomer (or tetramer-dimer) transition. Authors implicitly assume that this transition takes place before the formation of liquid droplets. How they know that the droplets are not promoted by EGFP? Does the EGFP itself form liquid or solid droplets?
Finally, do the EGFP-TTR fibers correspond to those obtained by using wild-type TTR?
Finally, if the conclusions of the paper are correct, I wonder if this can be considered a general mechanism of TTR fibrillation, if it is really relevant for the mechanism in vivo and if it could be useful for potential therapies. The (only) actual pharmacological treatment consists of compounds that binds to the TTR tetrameric central cavity, which have been demonstrated to be able to stabilize the tetramer and avoid the formation of monomers. This event precede the formation of droplets. Once the tetramer is dissociated, the fibers will form.
Minor points
Lines 57-59. TTR is … (CFS), respectively. Please rephrase, it is not clear to what the term respectively refers to.
Round 2
Reviewer 1 Report
The authors addressed the major points and improved the manuscript by considering the suggestions. The current version of the manuscript is suggested for publication.
Reviewer 2 Report
The authors have replied to all my comments, eventually performing new experiments, and I believe now the paper can be published.